# Dynamics and Molecular Interactions of GPI-Anchored CD59

**DOI:** 10.3390/toxins15070430

**Published:** 2023-06-30

**Authors:** Tomas B. Voisin, Emma C. Couves, Edward W. Tate, Doryen Bubeck

**Affiliations:** 1Department of Life Sciences, Sir Ernst Chain Building, Imperial College London, London SW7 2AZ, UK; 2Department of Chemistry, Molecular Sciences Research Hub, Imperial College London, London W12 0BZ, UK

**Keywords:** MACPF, pore-forming protein, complement, cholesterol-dependent cytolysins, CDC, membrane attack complex, GPI anchor

## Abstract

CD59 is a GPI-anchored cell surface receptor that serves as a gatekeeper to controlling pore formation. It is the only membrane-bound inhibitor of the complement membrane attack complex (MAC), an immune pore that can damage human cells. While CD59 blocks MAC pores, the receptor is co-opted by bacterial pore-forming proteins to target human cells. Recent structures of CD59 in complexes with binding partners showed dramatic differences in the orientation of its ectodomain relative to the membrane. Here, we show how GPI-anchored CD59 can satisfy this diversity in binding modes. We present a PyLipID analysis of coarse-grain molecular dynamics simulations of a CD59-inhibited MAC to reveal residues of complement proteins (C6:Y285, C6:R407 C6:K412, C7:F224, C8β:F202, C8β:K326) that likely interact with lipids. Using modules of the MDAnalysis package to investigate atomistic simulations of GPI-anchored CD59, we discover properties of CD59 that encode the flexibility necessary to bind both complement proteins and bacterial virulence factors.

## 1. Introduction

CD59 is a glycosylphosphatidylinositol (GPI)-anchored cellular receptor [1] that acts as a gatekeeper of protein pore formation. As an immune regulator, CD59 prevents pore formation by the complement membrane attack complex (MAC) [2,3]. It is the only membrane-bound inhibitor of the terminal pathway, abrogating membrane rupture by MAC [4] and further polymerization of the pore [5]. Ubiquitously expressed in human cells [2], the dysregulation of CD59 has devastating consequences for human disease including haemolytic anaemia [6] and cancer immune evasion [7,8]. In contrast to its role in preventing MAC pores, CD59 is co-opted by a subclass of bacterial pore-forming proteins to directly target human cells [9]. Intermedilysin (ILY), vaginolysin (VLY) and lectinolysin (LLY) are cholesterol-dependent cytolysins (CDCs), which also require CD59 to form pores [9,10,11]. Although these bacterial toxins can still bind cholesterol-rich membranes, CD59 determines the species specificity for these virulence factors [9] and coordinates oligomerization necessary for structural transitions en route to the final pore [12]. While there is extensive structural information on how CD59 engages MAC proteins [13] and CDCs [14,15], investigating the molecular interactions of GPI-anchored CD59 in a membrane environment will inform how CD59 satisfies its opposing roles in pore formation.

CDCs are bacterial virulence factors whose pore-forming domain shares structural homology with several human immune pores [16,17,18,19]. This domain, referred to as the membrane attack complex perforin (MACPF)/CDC fold, is characterized by an “L-shaped” four-stranded β-sheet flanked by two helical bundles [20]. During pore-formation, these helical bundles unfurl into transmembrane hairpins, which form a large β-barrel pore spanning the bilayer [19,21]. Despite a lack of sequence conservation in this domain, fundamental mechanisms of pore-formation are highly conserved. For both MAC and CDC pores, the process is initiated by the recruitment of soluble monomers to the membrane [22,23]. Monomeric proteins then oligomerize on the surface to trigger further conformational re-arrangement of regulatory regions, and ultimately, the helix-to-hairpin transition of transmembrane residues [24,25]. 

CD59 inhibits MAC pore formation through interactions with the MACPF/CDC domain of complement proteins [26,27]. CD59 blocks MAC at the first instance of membrane perforation by binding C8α [4,27]. Recent cryo-electron microscopy (cryoEM) studies have shown how CD59 re-directs pore-forming residues of the C8α MACPF [13]. CD59 binds C8α through a C-terminal β-strand of its central β-sheet, catching the cascading transmembrane residues and templating the newly formed C8α β-hairpin. The interface is further stabilized through an extensive hydrogen-bonding network across a β-sheet spanning both proteins. The C-terminal β-strand of CD59 bends the otherwise straight C8α hairpins, rendering them unable to pierce the lipid bilayer. Although this structure defines protein-protein interactions at the interface, it remained unclear which residues of a native GPI-anchored CD59 drive the orientation of its C-terminal β-strand to trap MAC.

Rather than binding CD59 through the MACPF/CDC domain, CDCs that require CD59 to make pores engage the receptor through their membrane binding domain [28]. CDCs have a modular domain architecture comprised of a pore-forming domain (Domains 1 and 3) and a three-stranded twisted β-sheet (Domain 2) that connects to the membrane-binding domain (Domain 4) [29]. The membrane binding domain is heavily β-stranded. Loops connecting the strands are responsible for lipid interactions [28] and an extended β-hairpin protruding from the core β-sheet engages the C-terminal β-strand of CD59 [14,15]. A comparison with the CD59-inihibited MAC complex showed that although the MAC and CDC binding interface of CD59 consists of a similar intermolecular β-sheet and involves residues common to both complexes [13,14,15,30,31], the orientation of CD59 relative to the membrane is dramatically different [13]. 

Protein-lipid interactions play a crucial role in the assembly and function of MAC and CDC pores. MAC deposition on the membrane causes changes in the rigidity of the bilayer [19], and we have recently shown that inhibited MAC assemblies induce local membrane thinning [13]. Although there is no lipid specificity for MAC to form pores, MAC co-localizes in cholesterol-containing microdomains on complement-activated mammalian cells [13,32]. Given that CDCs bind cholesterol and require this lipid for pore formation [28], the local membrane environment likely influences protein-protein interactions involving CD59. 

In this study, we set out to investigate the molecular basis of protein-lipid interactions that may influence CD59 function. Performing a PyLipID analysis [33] of our published coarse-grained molecular dynamics (MD) simulations [13], we discover residues of complement proteins that likely interact with either DOPC or cholesterol. By conducting an in-depth analysis of our atomistic simulations of GPI-anchored CD59 in a DOPC bilayer [13], we show that flexibility in the GPI anchor and protein moiety of CD59 defines its orientational landscape relative to the membrane. Our in silico analysis informs new testable hypotheses that will frame future biochemical and biophysical investigations of CD59 function.

## 2. Results

To understand the molecular basis for membrane distortion observed in the recently inhibited MAC structures, we analyzed coarse-grain simulations of the CD59-C5b8 complex in a lipid bilayer containing DOPC and cholesterol [17]. Unlike other structurally related pore-forming proteins [20,21], the luminal side of CD59-C5b8 does not repel lipids. Analysis with the PyLipID toolkit [22] revealed specific interactions between the lipids and complement proteins (Figure 1). Most of the longer-lived binding events involve DOPC, whereas cholesterol weakly interacts with the complex. Lipid-binding residues are found near the tips of the C6 and C7 membrane-interacting β-hairpins, consistent with the membrane-anchoring function of C7 [23,24]. C8β contains a stretch of lipid-binding residues located above the tip of the hairpins (Figure 1C). While the tip itself is hydrated as the protein spans the bilayer, neither water nor salt was observed to pass the lipid bilayer [17], in agreement with biochemical data showing CD59 blocks ion channel formation of C5b8 [25]. We observed several aromatic residues (C8β:F202, C7:F224 C6:Y285) together with arginine (C6:R407) and lysine (C8β:K326, C6:K412) residues within these lipid hotspots, consistent with their role in protein-lipid interactions at the membrane-water interface [26,27].

As MAC pores form within cholesterol-rich microdomains on human cells [17,28], we next explored if there was a preference for cholesterol or DOPC lipid interactions in our simulations. Our analysis shows that residues within the C6 β-hairpins exhibit longer dwell times for cholesterol when compared to other terminal pathway proteins (Figure 1B). By contrast, amino acids that interacted with DOPC in the simulation were distributed across C6, C7 and C8β. Taken together, our results show how specific residues within the pore-forming domain of complement proteins interact with the membrane to induce changes in their local lipid environment [11,17]. 

GPI-anchored proteins exhibit orientational preferences at the surface of the membrane [29]. To identify which residues influence the orientation of CD59 relative to the membrane, we next ran atomistic simulations of GPI-anchored CD59 in a DOPC bilayer [17]. We monitored the hydrogen bonds formed between the protein and DOPC, revealing that a cluster of residues located near the final β-strand of CD59 (CD59:Y61, CD59:Y62, CD59:K65 and CD59:K66) establishes contacts with the membrane (Figure 2). Among them, CD59:Y61 and CD59:Y62 are known to engage CDCs and MAC [15,16,17]. Mutating these residues leads to reduced activity of CD59, both in its role as a promoter and an inhibitor of pore formation [9,17,18,19]. These data suggest that in the absence of a binding partner, CD59 residues involved in biologically relevant interfaces can also interact with the membrane. Based on our MD simulations, together with our previous experimental structural data [15,17], we propose a model whereby solvent accessibility of the C-terminal β-strand of CD59 combined with its orientation relative to the cell surface define its role in mediating pore formation. 

Given the differences in CD59 orientations across the experimentally determined structures, we next sought to understand which amino acids within CD59 encode this flexibility. Crystal structures of soluble CD59 bound to bacterial pore-forming proteins show that CD59 forms an intermolecular anti-parallel β-sheet with the membrane-binding domain of these CDCs [15,16] (Figure 3A). Although CD59 forms a similar intramolecular β-sheet with complement protein C8α, the directionality of the C8α β-strands is such that CD59 undergoes a dramatic rotation to satisfy the anti-parallel arrangement (Figure 3A). These structures showed protein interfaces that defined the CD59 orientation; however, they did not include its native GPI anchor. To understand the molecular basis underpinning the flexibility of GPI-anchored CD59, we performed atomistic simulations within a DOPC membrane [17] and analyzed the position of CD59 relative to the lipid bilayer. We defined a vector between the backbone nitrogen atoms of the N- and C-terminal residues of CD59 and plotted the angle between this vector and one which is perpendicular to the membrane (Figure 3B). Our analysis shows that CD59 samples a wide range of angles, from 45 ° to 180 °, and encompasses both orientations previously observed in experimental structures (Figure 3B). We next calculated the root mean square fluctuation for each residue of CD59 from our atomistic simulations. Our analysis indicates that the most flexible regions lie within a series of loops clustered on one face of CD59 and the C-terminus, in agreement with the root mean square deviation of experimentally determined NMR structural ensembles of the soluble ectodomain of CD59 [30] (Figure 3C). The cluster of flexible loops is distal to a series of disulphide bonds that rigidify the structure. The highly flexible C-terminus directly precedes the GPI anchor. To explore the range of flexibility incurred by the linker, we defined a vector between the first and last sugar residues (inositol carbon C6 and mannose carbon C4) and plotted the distribution of angles sampled in our simulation between this vector and the one perpendicular to the membrane. Consistent with previous computational studies of GPI anchors [31], we observe a wide distribution of CD59 linker angles centered around 50°.

To understand the functional significance of CD59 flexibility, we investigated the conformational landscape of possible CD59 orientations. Specifically, we explored how the distribution and frequency of these orientations might influence interactions with binding partners. We extracted each frame from the atomistic simulations and aligned the protein moiety to a common reference in ChimeraX [32]. We used the resulting rotation matrices to calculate and plot the Euler angles of rotation about the x- and y-axes, describing the orientation of CD59 relative to the membrane (Figure 4A). CD59 samples a wide variety of orientations, consistent with the flexibility observed in Figure 3, but clusters in the orientation plot indicate that it preferentially adopts specific poses. To understand the functional significance of those clusters, the corresponding frames were visualized in ChimeraX [32] and aligned to the CD59-C5b8 and the CD59-ILY structures [16,17] (Figure 4B). One of the most populated orientation clusters places the final β-strand of CD59 in the correct position to engage the hairpins of C8α as they unfurl toward the membrane (Figure 4(B1)). A wide variety of orientations are compatible with CDC binding, including preferentially sampled poses (Figure 4(B3,B6)) and rarer orientations (Figure 4(B2,B4,B5)). This arises from the physical constraints imposed by both binding partners. C8α is part of a rigid membrane-attached complex, whereas CDCs engage CD59 as soluble monomers, providing them with more degrees of freedom. Finally, one of the most populated orientation clusters allows CD59 to interact with the membrane, where its final β-strand forms hydrogen bonds with lipids and is not available to bind either C8α or CDCs (Figure 4(B7)). This orientation is similar to the ‘flop-down’ conformation observed in previous simulations of GPI-anchored proteins [31,33]. Within this interface, CD59:Y61, Y62 and K66 form extensive interactions with the lipid bilayer. Previous mutagenesis studies have shown that both Y61 and Y62 of CD59 are essential for inhibiting MAC-mediated cell death [30,31]. CD59 residues Y62 and K66 form an integral part of the ILY-binding site [14], with mutations of CD59:Y62 influencing toxin binding and lytic activity [11].

## 3. Discussion

Complement activation occurs on the surface of lipid membranes and protein-lipid interactions are key to the function of MAC. MAC assembly alters the rigidity of the membrane [19] and inhibited MAC complexes induce a local thinning of the bilayer [13]. Here, we have shown with coarse-grain MD simulations that specific residues within the β-hairpins of complement proteins directly bind lipids, at a stage where the membrane is not permeabilised by MAC [4,13]. While both DOPC and cholesterol interact with MAC, the binding sites and residence times differ for both lipids, suggesting a level of lipid specificity. Although pore formation by MAC is not dependent on the lipid composition of the target membrane, lipid-specific interactions may lead to different effects on the physical properties of the membrane. Indeed, monoclonal antibody therapeutics that activate complement target cholesterol-containing rafts [32] and our previous work demonstrated that MAC clustering on the plasma membrane depends on cholesterol [13]. Our results provide a molecular basis for these protein-lipid interactions in the context of MAC formation and inhibition. 

CD59 is the gatekeeper of membrane rupture on the surface of human cells, where it acts as an inhibitor of MAC [35] and a promoter of pore formation by CDCs [9,10,11]. Both activities rely on protein-protein interactions involving the final β-strand of CD59, but it must adopt dramatically different orientations relative to the membrane to engage either type of binding partner. Our atomistic simulations reveal that flexibility in the C-terminus of the CD59 ectodomain and in the sugar moiety of the GPI anchor allow CD59 to adopt a wide variety of orientations. Mapping those orientations (Figure 4A) reveals that CD59 preferentially samples orientations where its final β-strand is available to engage MAC or CDCs. Our results differ from previously published data from simulations of CD59 in an erythrocyte-like membrane, where they mainly sampled the membrane-interacting ‘flop-down’ conformation [36]. This is likely a result of different lipid compositions, suggesting that the local lipid environment of CD59 plays a role in its function. 

## 4. Materials and Methods

The full methods used for preparing and running the simulations analysed in this study were published in [13]. Briefly, a coarse-grain model of the CD59-C5b8 complex was placed in a 7:3 (molar ratio) DOPC:cholesterol membrane and hydrated in polarisable water [37] with 0.15 M NaCl. Three independent replicates (2 µs each) were simulated with the Martini 2 forcefield. Atomistic GPI-anchored CD59 was modelled in a DOPC membrane using the CHARMM-GUI webserver [38,39] and hydrated in TIP3 [40] water with 0.15 M NaCl. Three independent replicates (500 ns each) were simulated with the CHARMM36m forcefield [41]. All simulations were run with Gromacs 2021.3 [42].

### 4.1. Analysis of Coarse-Grain CD59-C5b8 Simulations

Lipid binding sites within the coarse-grain CD59-C5b8 complex were identified using the PyLipID Python 3 toolkit [33] with a lower cut-off of 0.55 nm and an upper cut-off of 1.0 nm. The lipid residence time of each residue was plotted as a function of its occupancy, separately for DOPC and cholesterol. 

### 4.2. Analysis of Atomistic CD59 Interactions with the Membrane

Potential hydrogen bond partners between atomistic GPI-anchored CD59 and the membrane were identified with the HydrogenBondAnalysis module of MDAnalysis [43,44] in Python 3. The number of hydrogen bonds formed by each potential donor on CD59 with DOPC was counted in every frame from the three replicates and plotted as a function of residue number. 

### 4.3. Analysis of CD59 Flexibility

To measure flexibility in the atomistic simulations of CD59, the angle between the z-axis of the box (taken as an approximation of the normal of the membrane plane) and a vector connecting the backbone nitrogen atoms of CD59:L1 (N-terminus) and CD59:N77 (C-terminus) was tracked over the course of the simulations with the gangle tool in Gromacs [42]. The angle between the z-axis and a vector connecting carbon C6 of the inositol residue and carbon C4 of the protein-proximal mannose residue within the GPI anchor was similarly analysed. The per-residue Cα root mean square fluctuation of the protein was calculated over the three replicates with the MDAnalysis.analysis.rms module of MDAnalysis [43,44,45,46] and compared to the root mean square deviation of the NMR ensemble of soluble CD59 (PDB 1CDR) [47] calculated in UCSF Chimera v. 1.4 [48].

### 4.4. CD59 Orientation Plot

The orientation sampling of CD59 relative to the membrane was calculated by aligning each frame of the simulations with the same reference. The final configuration after equilibration of one of the replicates was used at the reference. The alignment was performed with the align command of UCSF ChimeraX v. 0.6 [49] and the rotation matrix was extracted for each frame. Euler angles were calculated from the rotation matrices and plotted to obtain an orientation landscape of GPI-anchored CD59. 

A rotation matrix R generated by the align command of UCSF ChimeraX v. 1.6 [49] takes the following form (after exclusion of the translation terms):$$R=\left[\begin{array}{ccc} R_{11} & R_{12} & R_{13}\\ R_{21} & R_{22} & R_{23}\\ R_{31} & R_{32} & R_{33} \end{array}\right]$$

The angles of rotation θ (about the y-axis) and ψ (about the x-axis) can be calculated with the following two equations, provided that $\cos\left(\theta \right)\neq 0$:$$\theta =-\sin^{-1}\left(R_{31}\right)$$
$$\psi =atan2\left(\frac{R_{32}}{\cos\left(\theta \right)},\frac{R_{33}}{\cos\left(\theta \right)}\right)$$

The angle θ accepts another solution, which is shifted by π compared to the one above. As a result, a single solution is enough to generate a complete orientation plot. The z-axis was taken as an approximation of the normal of the membrane and only the angles of rotation about the x- and y-axes were included in the analysis. The results were plotted using the Seaborn package v. 0.12.2 [34] in Python 3.

## Figures and Tables

**Figure 1 toxins-15-00430-f001:**
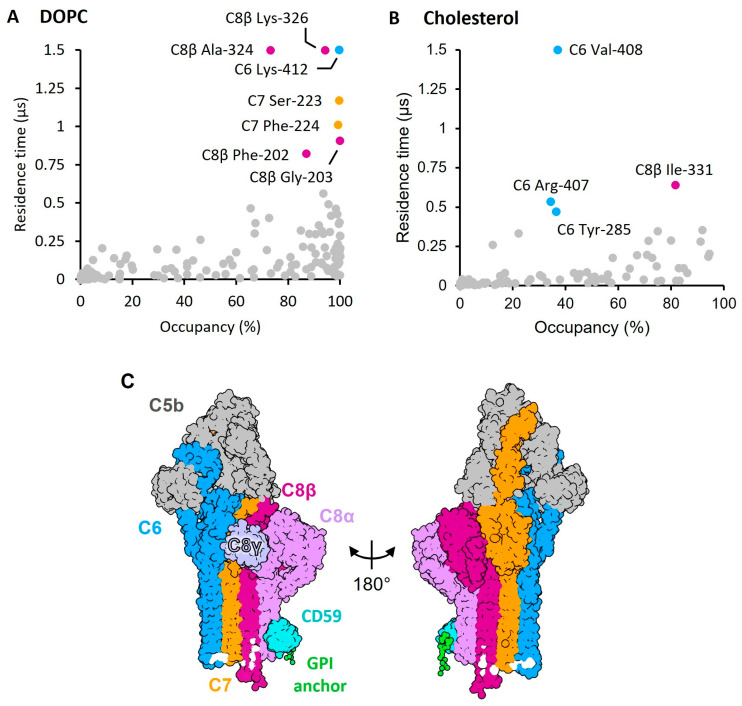
Mapping protein-lipid interactions on the CD59-C5b8 complex. Potential lipid binding sites highlighted by PyLipID [33] from coarse-grain MD simulations of CD59-C5b8 in a membrane containing DOPC and cholesterol. (**A**) Occupancy and residence time of DOPC for each residue. (**B**) Occupancy and residence time of cholesterol for each residue. As cholesterol interactions are weaker, the cut-off residence time and occupancy values for a hotspot were lower than for DOPC. Potential binding sites in panels A and B are coloured according to MAC protein component (C5b grey; C6 blue; C7 orange, C8α light pink; C8β magenta; C8γ light purple; CD59 cyan). Results averaged over three independent replicates. (**C**) Mapping of the potential hotspots in panels A and B (highlighted in white) on the coarse-grained model generated from PDB 8B0F, coloured according to complement component.

**Figure 2 toxins-15-00430-f002:**
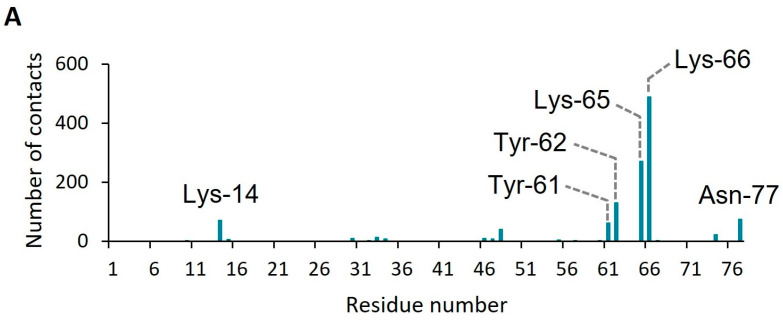
CD59 residues involved in hydrogen bonds with DOPC. (**A**) Average number of hydrogen bonds established with DOPC by each CD59 residue over the course of three independent simulations. (**B**) Structure of CD59 (PDB 1CDR) with residues highlighted in panel A shown as sticks.

**Figure 3 toxins-15-00430-f003:**
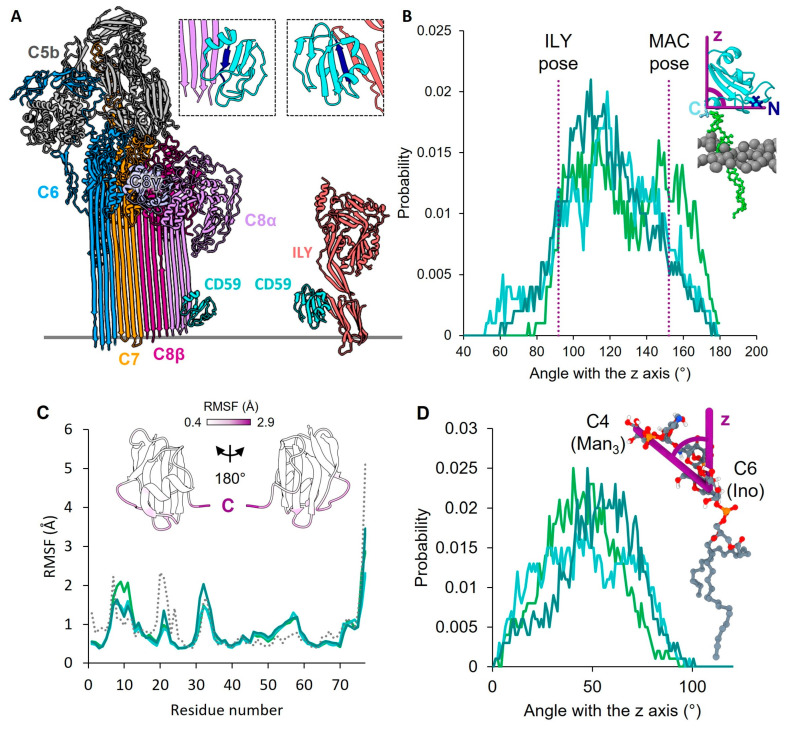
CD59 flexibility in atomistic MD simulations. (**A**) Cryo-electron microscopy structure of the CD59-C5b8 complex (PDB 8B0F) (left panel) and crystal structure of CD59 (cyan) in complex with the bacterial pore-forming protein ILY (red) (PDB 5IMT) (right panel). The plane of the membrane is schematically shown by the grey line. Inset: close-up views of the CD59-C5b8 (left) and CD59-ILY (right) interfaces, with the final β-strand of CD59 shown in dark blue. Complement proteins coloured as in Figure 1. (**B**) Distribution of the angle measured between the vector perpendicular to the membrane (z-axis) and a vector connecting the C- and N-termini (passing through their backbone nitrogen atoms). Each solid line shows the distribution over the course of a replicate. The values of the angle corresponding to the ILY-bound and MAC-bound orientations are shown by the vertical purple dotted lines. Inset for panel B: representation of the angle measured (purple). Lipid headgroups are in grey. (**C**) Per-residue root mean square fluctuation (RMSF) of CD59 in the atomistic simulations (solid lines show the per-residue Cα RMSF for each replicate). The root mean square deviation derived from the NMR ensemble (PDB 1CDR) is shown in dotted grey for comparison. Inset for panel C: CD59 coloured by RMSF averaged over the independent triplicates. The C-terminus is indicated by the letter C. (**D**) Distribution of the angle between the vector perpendicular to the membrane (z-axis) and the vector connecting carbon C6 of the inositol residue and carbon C4 of the protein-proximal mannose residue within the GPI anchor. Inset for panel D: representation of the angle measured (purple). The different shades of green in panels B, C and D correspond to independent triplicates.

**Figure 4 toxins-15-00430-f004:**
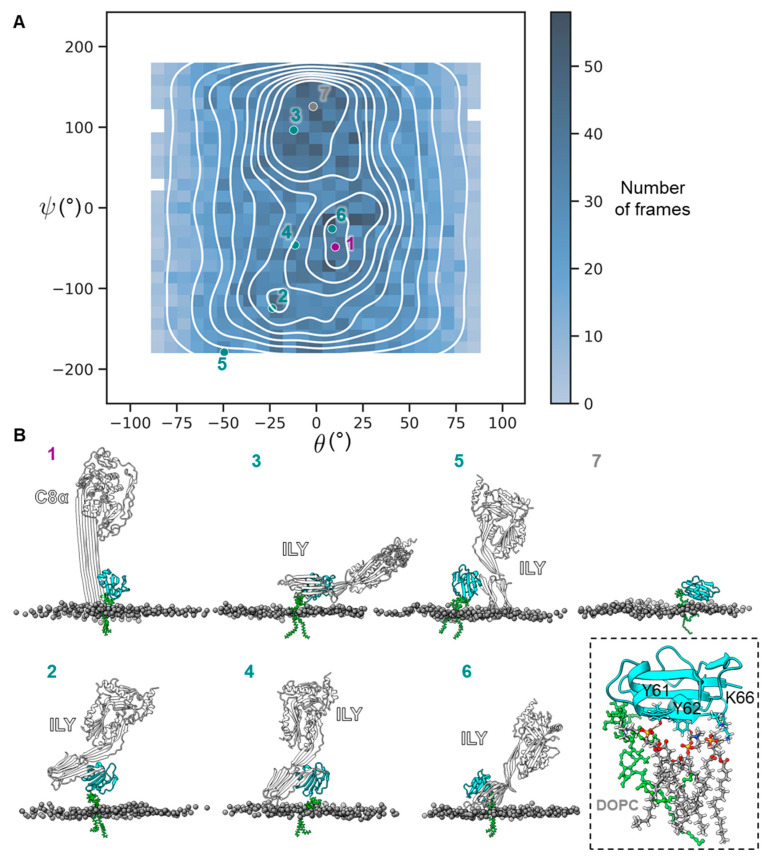
Orientations of CD59 relative to the membrane. (**A**) Angular distribution plot of CD59 θ and ψ rotation angles (around the y- and x-axes, respectively) from all trajectories compared to the final step of equilibration for one of the replicates. The lines represent the contours of a kernel density estimation performed by Seaborn [34]. Selected orientations are shown by numbered dots. Purple: orientation compatible with C8α binding; turquoise: orientations compatible with ILY binding; grey: orientation compatible with hydrogen-bond formation with the membrane. (**B**) Selected CD59 orientations from panel A. **1**: The CD59-C5b8 complex (PDB 8B0F) was aligned to the frame (2.9 Å RMSD) and C8α is shown in white. **2**–**6**: The CD59-ILY (PDB 5IMT) complex was aligned to each frame (respective RMSD of 1.3 Å, 1.3 Å, 1.4 Å, 1.6 Å and 1.9 Å) and ILY is shown in white. **7**: CD59 cannot accommodate C8α or ILY, but instead forms hydrogen bonds with the membrane. Inset: Close-up view of CD59 with the DOPC molecules engaging in hydrogen bonds with residues CD59:Y61, CD59:Y62 and CD59:K66 shown as sticks.

## Data Availability

Initial and final configurations for the MD simulations analysed in this study are publicly available as source data associated with the publication https://doi.org/10.1038/s41467-023-36441-z. Data files used in the plots for this study are uploaded to Zenodo database (https://doi.org/10.5281/zenodo.8029612).

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
