# Peer review of "Dynamics and Molecular Interactions of GPI-Anchored CD59"

_toxins, 2023, doi:10.3390/toxins15070430_

Round 1

Reviewer 1 Report

Through dynamic simulation, this research describes the interaction of the complement membrane attack complex (MAC) with lipids of the plasmatic membrane, particularly with cholesterol, as well as the interactions of CD59, describing the flexibility it can have in this molecule. It is a manuscript with interesting contributions to lipid-protein interaction, specifically the different modes of interaction that CD59 can have. However, it is essential to address some points,

Modifying the title and correcting the syntax is suggested; it is confusing. Specify the type of membrane or type of lipids.

In the abstract, it is missing to specify the program used for the simulation analyses; this is part of the method. Mention the relevant amino acid residues in the interaction with lipids.

Key contribution: the central research question focuses on CD59 interactions, emphasizing results that answer the central question.

Lines 40-45 missing references.

At the bottom of Figure 1, indicate the PDB code of the structure considered for the models and the RMSD values.

Figure 3. In the foot of the figure indicate the description of figure 3C

Figure 4 highlights the location of the most critical residues in the interaction.

During the simulation, why was NaCl not considered?

The position of CD59 in Figure 4B-7; which amino acid residues are involved in this interaction? Is there experimental evidence that these residues have been mutated? If so, what effects does it have on the interaction of CD59 with MAC and the membrane?

The manuscript lacks clarity in writing. 

Author Response

Through dynamic simulation, this research describes the interaction of the complement membrane attack complex (MAC) with lipids of the plasmatic membrane, particularly with cholesterol, as well as the interactions of CD59, describing the flexibility it can have in this molecule. It is a manuscript with interesting contributions to lipid-protein interaction, specifically the different modes of interaction that CD59 can have. However, it is essential to address some points,

Modifying the title and correcting the syntax is suggested; it is confusing. Specify the type of membrane or type of lipids.

We analysed CD59 within the context of 2 different MD simulations: 1) a course grained simulation that was performed in a DOPC/Cholesterol membrane and 2) and atomistic simulation performed in a DOPC membrane. Given these differences, we feel it is too detailed to include this information in the title. We have modified the title to improve the grammatical syntax and to reflect the analyses undertaken. The new title is:

“Dynamics and molecular interactions of GPI-anchored CD59.”

In the abstract, it is missing to specify the program used for the simulation analyses; this is part of the method. Mention the relevant amino acid residues in the interaction with lipids.

We have now included the names of the programs/tools used in the analysis of both the course-grained and atomistic simulations. In the abstract, we have now included the amino acid residues that likely interact with lipids highlighted in the Results section.

“We present a PyLipID analysis of course-grain molecular dynamics simulations of a CD59-inhibited MAC to reveal residues of complement proteins (C6:Y285, C6:R407 C6:K412, C7:F224, C8β:F202, C8β:K326) that likely interact with lipids. Using modules of the MDAnalysis package to investigate atomistic simulations of GPI-anchored CD59, we discover properties of CD59 that encode the flexibility necessary to bind both complement proteins and bacterial virulence factors.”

Key contribution: the central research question focuses on CD59 interactions, emphasizing results that answer the central question.

We have changed the Key contributions section to reflect this:

“In this manuscript we explore molecular dynamics simulations of GPI-anchored CD59 to understand how CD59 encodes the flexibility necessary for its two roles in complement regulation and in host-pathogen interactions.”

Lines 40-45 missing references.

The introduction has been extensively re-written and appropriately referenced.

At the bottom of Figure 1, indicate the PDB code of the structure considered for the models and the RMSD values.

The PDB code is now included in the legend for Figure 1 and the RMSD values are included in the updated legend for Figure 4.

Figure 3. In the foot of the figure indicate the description of figure 3C

There was an error in the figure 3 legend callouts. This is now fixed. The inset for panel 3C is now more clearly described in the legend.

Figure 4 highlights the location of the most critical residues in the interaction. During the simulation, why was NaCl not considered?

NaCl and water were both tracked during the simulation. Neither crossed the membrane. We have now made this clearer in the Results and Methods sections.

“While the tip itself is hydrated as the protein spans the bilayer, neither water or salt was observed to pass the lipid bilayer [17], in agreement with biochemical data showing CD59 blocks ion channel formation of C5b8 [25].”

“Briefly, a coarse-grain model of the CD59-C5b8 complex was placed in a 7:3 (molar ratio) DOPC:cholesterol membrane and hydrated in polarisable water [37] with 0.15 M NaCl. Three independent replicates (2 µs each) were simulated with the Martini 2 forcefield. Atomistic GPI-anchored CD59 was modelled in a DOPC membrane using the CHARMM-GUI webserver [38,39] and hydrated in TIP3 [40] water with 0.15 M NaCl. Three independent replicates (500 ns each) were simulated with the CHARMM36m forcefield [41]. All simulations were run with Gromacs 2021.3 [42]”

The position of CD59 in Figure 4B-7; which amino acid residues are involved in this interaction? Is there experimental evidence that these residues have been mutated? If so, what effects does it have on the interaction of CD59 with MAC and the membrane?

We have now labelled residues involved in this interaction with the membrane (CD59: Y61, Y62 and K66) in a revised Figure 4 legend and discussed the effect mutation of these residues has on CD59 function.

“Within this interface, CD59:Y61, Y62 and K66 form extensive interactions with the lipid bilayer. Previous mutagenesis studies have shown that both Y61 and Y62 of CD59 are essential for inhibiting MAC-mediated cell death [30,31]. CD59 residues Y62 and K66 form an integral part of the ILY-binding site [14], with mutation of CD59:Y62 influencing toxin binding and lytic activity [11].”

The manuscript lacks clarity in writing. 

The introduction has been re-written to improve the clarity of the writing.

Reviewer 2 Report

Although an in silico analysis may be interesting due to the proposals that arise from it, it is no more than a hypothesis, which requires validation to be of full interest and value. Without a doubt, this model of CD59 and its interactions are of high value and interest, but without an experimental analysis that verifies the model, I do not see the value of the model.

Author Response

Although an in silico analysis may be interesting due to the proposals that arise from it, it is no more than a hypothesis, which requires validation to be of full interest and value. Without a doubt, this model of CD59 and its interactions are of high value and interest, but without an experimental analysis that verifies the model, I do not see the value of the model.

We have mentioned in the last paragraph of the introduction that our in silico analysis informs new testable hypotheses that will frame future biochemical and biophysical investigations of CD59 function. While we agree with the reviewer that follow-up experimental analysis will further advance the field, that is beyond the scope of this study.

Reviewer 3 Report

In this manuscript the authors aim to understand at the molecular level how CD59 interacts with MAC to limit immune pore formation and how it interacts with bacterial pore-forming toxins to promote pore-formation. Despite this manuscript only reports data obtained in atomistic simulations, the analysis presented has a real added value to our current knowledge.

The manuscript is well written and the content should be of the interest of researchers working on the field.

Author Response

In this manuscript the authors aim to understand at the molecular level how CD59 interacts with MAC to limit immune pore formation and how it interacts with bacterial pore-forming toxins to promote pore-formation. Despite this manuscript only reports data obtained in atomistic simulations, the analysis presented has a real added value to our current knowledge.

The manuscript is well written and the content should be of the interest of researchers working on the field.

We thank the reviewer for their appreciation of our work.

Round 2

Reviewer 2 Report

no more comments